

# The rise of Chrome

Jonathan Tamary and Dror G. Feitelson

The Rachel and Selim Benin School of Computer Science and Engineering, The Hebrew University of Jerusalem, Israel

## ABSTRACT

Since Chrome's initial release in 2008 it has grown in market share, and now controls roughly half of the desktop browsers market. In contrast with Internet Explorer, the previous dominant browser, this was not achieved by marketing practices such as bundling the browser with a pre-loaded operating system. This raises the question of how Chrome achieved this remarkable feat, while other browsers such as Firefox and Opera were left behind. We show that both the performance of Chrome and its conformance with relevant standards are typically better than those of the two main contending browsers, Internet Explorer and Firefox. In addition, based on a survey of the importance of 25 major features, Chrome product managers seem to have made somewhat better decisions in selecting where to put effort. Thus the rise of Chrome is consistent with technical superiority over the competition.

## INTRODUCTION

The most notable use of the Internet is the World Wide Web (WWW). The web was created by Tim Berners-Lee and his colleagues at CERN (The European Organization for Nuclear Research) in 1989. In order to consume information from the web, one must use a web browser to view web pages. The first web browser (which was in fact named WorldWideWeb) was developed at CERN as part of the WWW project (http://www.w3.org/People/Berners-Lee/WorldWideWeb.html). But the first popular browser, which set the growth of the web in motion towards the wide use we see today, was Mosaic, which was developed by Marc Andreessen and Eric Bina at the National Center for Supercomputing Applications (NCSA) in 1993 (*Vetter, Spell & Ward, 1994*).

The open nature of the web makes it possible for different browsers to co-exist, possibly providing different features, user interfaces, and operating system support. Over the years different browsers have competed for the user's choice. The competition between browsers has led to several "browser wars"—periods of fierce competition between different web browsers that are characterized by technological innovation and aggressive marketing, typically leading to the eventual dominance of one browser and the fall of another. In recent years we have witnessed such a shift (albeit somewhat protracted) from Microsoft's Internet Explorer to Google's Chrome.

The reasons for this shift are most probably a mix of technical reasons and marketing reasons. Our goal is to explore the technical aspects, and see whether they can explain the growing popularity of Chrome. In particular, we wanted to assess the technical quality

Corresponding author
Dror G. Feitelson, feit@cs.huji.ac.il

of chrome and compare it with the quality of its rivals. To do so we downloaded all the versions of Chrome, Firefox, and Internet Explorer that were released over a period of five years, and compared them using a set of benchmarks which together provide a rather comprehensive coverage of browser functionality and features. As far as we know our work is by far the widest study of its kind.

In a nutshell, we find that Chrome is indeed technically superior to other browsers according to most commonly-used benchmarks, and has maintained this superiority throughout its existence. Also, based on a survey of 254 users, the features pioneered by Chrome ahead of its competitors tend to be those that the users consider more important. Thus Chrome's rise to dominance is consistent with technical superiority. However, one cannot rule out the large effect of the Google brand and the marketing effort that was invested as factors that contributed greatly to the realization of Chrome's technical potential.

## THE BROWSERS LANDSCAPE

### Browsers history

Not long after the release of the Mosaic web browser in 1993 it became the most common web browser, keeping its position until the end of 1994. The factors contributing to Mosaic's popularity were inline graphics, which showed text and graphics on the same page, and popularizing the point and click method of surfing. Moreover, it was the first browser to be cross-platform including Windows and Macintosh ports. Amazingly, by the end of 1995 its popularity plummeted to only 5% of the web browser market (*Berghel, 1998*). This collapse in Mosaic's popularity was concurrent to the rapid rise of Netscape Navigator which was released in December 1994 and managed in less than two years to reach around 80% market share (different sources cite somewhat different numbers).

Several factors are believed to have caused the fast adoption of Netscape by users. First, it was a natural followup of Mosaic as it was developed by the same people. Second, Netscape introduced many technological innovations such as on-the-fly page rendering, JavaScript, cookies, and Java applets (*Berghel, 1998*). Third, Netscape introduced new approaches to testing and distribution of web browsers by releasing frequent beta versions to users in order to test them and get feedback (*Yoffie & Cusumano, 1998*).

Netscape's popularity peeked in 1996 when it held around 80% market share. But in August 1995 Microsoft released the first version of Internet Explorer based on an NCSA Mosaic license. A year later, in August 1996, with the release of Internet Explorer 3, a browser war was on. By August 1999 Internet Explorer enjoyed 76% market share (*Windrum, 2004*).

During this browser war it seems that Internet Explorer did not have any technological advantage over Netscape, and even might have been inferior. Therefore, other reasons are needed to explain Internet Explorer's success. One reason was that Netscape's cross platform development wasn't economical: instead of focusing on one dominant platform (Windows) it had approximately 20 platforms which caused a loss of focus. Meanwhile, Microsoft focused on only one platform. Second, Microsoft bundled Internet Explorer with Windows without a charge, and as Windows dominated the desktop operating systems market Explorer was immediately available to the majority of users without any

 

effort on their part. In an antitrust investigation in the US, Microsoft was found guilty of abusing its monopoly in the operating systems market by bundling Internet Explorer with Windows for free. Lewis describes this as follows (*Lewis, 1999*):

> "Adding Internet Explorer to Windows 95 and calling it Windows 98 is innovation in Gates' terminology, but it is monopolizing according to DOJ."

Settling the antitrust case took several years (October 1997–November 2002), during which Internet Explorer deposed Netscape as the most popular browser. And once Internet Explorer was entrenched, its market share grew even more due to a positive feedback effect. The standard tags used in HTML (Hyper-Text Markup Language, in which web pages are written) are defined by the W3C (World Wide Web Consortium). However, both Microsoft and Netscape extended the HTML standard with their own special tags, thus creating two competing sets of HTML tags and behaviors. Web developers with limited resources then had to choose one of these tag sets, and as Internet Explorer usage grew they opted to use Internet Explorer's extensions, thereby making it ever more preferable over Netscape (*Phillips, 1998*). The dominance of Internet Explorer was so strong that Microsoft didn't bother to release a major version of Internet Explorer from 2001 until 2006, making do with a Service Pack for Internet Explorer 6 as part of a Windows XP Service Pack.

Up to this point browsers were proprietary software, even if distributed for free. But with the collapse of Netscape's market share, Netscape released its Netscape Communicator 5.0 source code in March 1998 for community involvement in the development via mozilla.org (cached version of http://www.netscape.com/newsref/pr/newsrelease591. html at http://xml.coverpages.org/netscapeCode980401.html). The Mozilla Suite was created based on this source code release. However, the development continued to be influenced by Netscape Communications Corporation, which had been acquired by AOL. David Hyatt, Joe Hewitt, and Blake Ross were not pleased with the alliance of Mozilla with Netscape, which was hurting Mozilla independence and more importantly led to feature creep and bloat. So in mid 2001 they created an experimental branch of the Mozilla Suite, which kept the user interface but reimplementing the backend from scratch (https: //web.archive.org/web/20110623034401/http://weblogs.mozillazine.org/ben/archives/ 009698.html). This branch became the open source Firefox browser, and on August 9, 2003 Mozilla released a revised road map that reflected a shift from the Mozilla Suite to Firefox (http://www-archive.mozilla.org/roadmap/roadmap-02-Apr-2003.html). Firefox was finally released on November 9, 2004 (http://website-archive.mozilla.org/www. mozilla.org/firefox_releasenotes/en-US/firefox/releases/1.0.html). Later, in March 2011, Mozilla moved to rapid development with a 16-week cycle and then a 6-week cycle (http:// mozilla.github.io/process-releases/draft/development_overview/, http://blog.mozilla.org/ channels/2011/07/18/every-six-weeks/).

Firefox's market share grew slowly, and by the end of 2009 it managed to wrestle away up to 30% from Internet Explorer. But by this time a new contender had arrived. Google released Chrome 1.0 on September 2, 2008. Concurrently, Google released most of the browser's source code as an open source project named Chromium thus establishing an

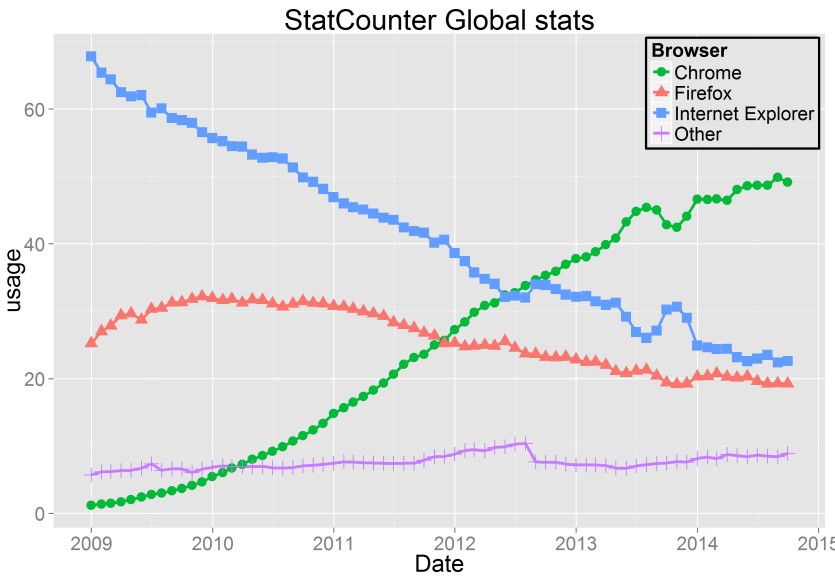

**Figure 1** **Browsers usage data from StatCounter.com.**

open source community. The main reason was the belief that a strong community will help improve Chrome (http://blog.chromium.org/2008/09/welcome-to-chromium_02.html). Additional reasons were to help drive the web forward as other open source projects like Firefox and WebKit did, and enabling people to create their own projects based on Chromium. As of today the development of Chrome is based on the development of stable releases of Chromium, and the two browsers are identical in most aspects (https://code.google.com/p/chromium/wiki/ChromiumBrowserVsGoogleChrome). However, it is important to distinguish Chrome from Chromium, as Chrome has several features that are absent from Chromium such as a non-free PDF viewer. Chrome and Chromium moved to a rapid development strategy in mid 2010 (http://blog.chromium.org/2010/07/release-early-release-often.html).

## Browsers usage statistics

In the six years since its release Chrome has dethroned Internet Explorer, and Firefox's market share has also decreased, as shown in Fig. 1. Data for such statistics is obtained as follows. Browser usage can be tracked using a counter embedded in the HTML source code of web sites. The counting is implemented using a request to a counting service, enabling the counting service to also extract the browser information from the request and to use it to tabulate browser usage statistics. The data shown is from one of these services, StatCounter.com (http://gs.statcounter.com/).

There are two main methods to interpret web browsers usage. The first method is to measure how many page loads came from each type of browser in a certain period of time. The second method counts how many unique clients (installations) were active in a certain period of time. Therefore, if a user visits 10 web pages, the first method will count these visits as 10 uses of the browser, while in the second method will count them as one

user. Since the two methods measure different parameters their results may differ. The first method favors browsers that are used by heavy users, while the second method just counts the number of unique users without taking their activity into account, which may be a drawback if we consider users who use the web extensively to be more important. Moreover, identifying unique users is non-trivial and requires manipulating the raw data. We therefore use the raw counts data, and specifically the data for desktop browsers only not including mobiles and tablets. (The nick in the graphs at August 2012 represents the beginning of collecting data about tablets separately.)

As shown in the graph, Chrome's market share has risen consistently over the years, largely at the expense of Internet Explorer. As of January 2015, Chrome was responsible for 51.7% of the page loads while Internet Explorer was responsible for 21.1%, Firefox for 18.7%, and other browsers for 8.4%.

Note that in fact any site can track the distribution of browser usage among the users who access that site. Such tracking may lead to different results if the visitors to a certain site prefer a certain browser. For example, w3schools.com (a site devoted to tutorials about web development) also publishes data about browser usage. Their results for January 2015 are that 61.9% use Chrome, 23.4% use Firefox, and only 7.8% use Internet Explorer (http://www.w3schools.com/browsers/browsers_stats.asp). This probably reflects a biased user population of web developers who tend to work on Linux platforms rather than on Windows. At the opposite extreme, netmarketshare.com claims that only 23% of the market uses Chrome, while fully 58% still use Internet Explorer (these figures are again for January 2015) (http://netmarketshare.com/). There is a danger that the StatCounter data is also biased, but it is thought to provide a good reflection of common usage by the public on popular web sites, because its data is based on counters embedded in many different sites. Further justification is given in the threats to validity section.

## RESEARCH QUESTIONS

Despite possible differences in usage statistics, it is clear that Chrome is now a dominant player in the web browser market. The question is how this dominance was achieved, and in particular whether it is justified from the technical point of view. We divide this into the following specific questions:

1. Is Chrome technically superior to its competitors? Specifically,
   (a) Is the performance of Chrome superior to that of its competitors as measured by commonly accepted browser performance benchmarks?
   (b) Is the start-up time of Chrome competitive with the start up times of its competitors?
   (c) Does Chrome conform to web standards better than its competitors as measured by commonly accepted browser conformance benchmarks?
2. Given that the browser market is not static and web usage continues to evolve,
   (a) Did Chrome introduce features earlier than its competitors?
   (b) Were the features that Chrome introduced first more important than those introduced by its competitors?

To answer these questions we tested the three major browsers which together account for 91% of the market share. Thus we did not initially test Opera and Safari, whose market share is very low; Safari is also less relevant as it is tightly linked to the Mac OS X platform, so it does not compete with Chrome for most users.[1] The question regarding the release of important features earlier also involved a wide user survey to assess the relative importance of different features. Note that the performance and conformance evaluations are not tied to one point in time, but rather they are evaluated over the whole period when chrome achieved its rise in market share. As a result we also found interesting information about the consistency (and sometimes inconsistency) of browser benchmarks which was not anticipated in advance.

## TECHNICAL PERFORMANCE

In this section we present the methodology and results pertaining to answering research question (1), namely the relative performance of Chrome and the competing browsers.

### Experimental design

Timing is important to web page designers, because it affects the user experience (*Sounders, 2008*). But the precise definitions of performance metrics are complicated to pin down (*Meenan, 2013*). As a result quite a few different benchmarks have been designed and implemented over the years. Instead of proposing yet another benchmark, we used several of the more widely accepted and commonly used benchmarks to evaluate the technical performance of the different browsers, selecting a set which cover a wide range of functionalities.

These benchmarks are divided to two categories. The first category is performance, and tests the performance of different aspects of the browsers. This included general browser performance, aspects of JavaScript processing, and in particular support for the HTML5 <canvas> tag. The second category is conformance, and tests the conformance of the different browsers to common standards such as the HTML5 and CSS3 standards. Note, however, that the tests typically check only that elements of the standard are recognized, and not the quality of the implementation. This is because assessing the quality may be subjective and depend on graphical appearance. In addition we implemented our own methodology for measuring startup times, as this was not covered by the available benchmarks. The benchmarks and their response variables are listed in Table 1.

Note that the benchmarks do not include low-level issues such as memory usage. The reason is that these benchmarks are not intended to characterize the interaction between the browser and the underlying hardware platform, but rather the interaction of the browser with the user. We selected these benchmarks because market share depends on users who are more influenced by general performance and features, not by details of hardware utilization.

In order to assess the technical performance of the competing browsers during the period when Chrome gained its market leadership, we measured all the releases of these browsers using all the benchmarks. Specifically, the measurements covered all Chrome versions from 1 to 31, all Firefox versions from 3 to 26, and all Internet Explorer versions

[1] A Windows version was available for only several years and then discontinued.

**Table 1** The benchmarks used in our study and their response variables.

| Benchmark | Type | Content | Response |
|---|---|---|---|
| SunSpider | Performance | Javascript tasks | Time |
| BrowserMark | Performance | General browser performance | Score |
| CanvasMark | Performance | \<canvas\> tag | Score |
| PeaceKeeper | Performance | Javascript tasks | Score |
| Start-up times | Performance | Cold startup time | Time |
| HTML5 Compliance | Conformance | HTML standard | Score |
| CSS3 Test | Conformance | CSS standard | Score |
| Browserscope Security | Conformance | Security-enhancing features | Tests passed |

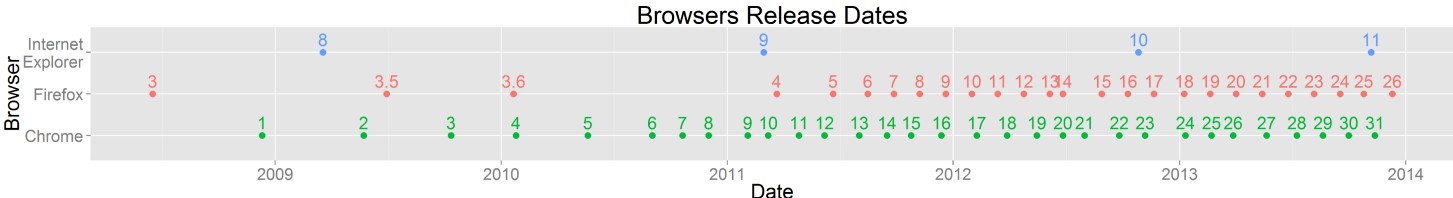

**Figure 2** The release date of each version tested.

from 8 to 11, meaning all the browser versions in a five year span starting in mid 2008 until the end of 2013 (Fig. 2). This is the period from the first release of Chrome until it achieved around 50% market share.

## Execution of measurements

The measurements were conducted on two identical Core i5 computers (Lenovo ThinkCentre M Series with i5-3470 CPUs running at 3.20 GHz) with 4 GB RAM each, and Windows 7 Professional SP1 operating systems. One machine ran Windows 7 32 bit and the other ran Windows 7 64 bit. The browser versions used on the 32 bit system were Chrome 1–12, Firefox 3–5, and Internet Explorer 8–9, i.e., all the browsers released up to May 2011. The browsers versions used on the 64 bit system were Chrome 13–31, Firefox 6–26, and Internet Explorer 10–11. The versions were divided between the machines since we encountered some compatibility issues with earlier versions of Chrome on Windows 7 64 bit. Moreover, it makes sense to switch to 64 bit along the way to reflect the growing adoption of 64-bit systems.

To make sure that the measurements are consistent between 32 bit and 64 bit operating systems and eliminate operating system bias we have checked a third of the browser versions on both systems, focusing on versions with a six months release gap. Examples for two of the benchmarks are shown in Fig. 3. SunSpider 1.0.2 is an example of a relatively big difference between the results on the two platforms (which is still quite small), and PeaceKeeper is an example of a very small difference. In general we did not see any dramatic differences between the platforms. We therefore do not present any more results about such comparisons.

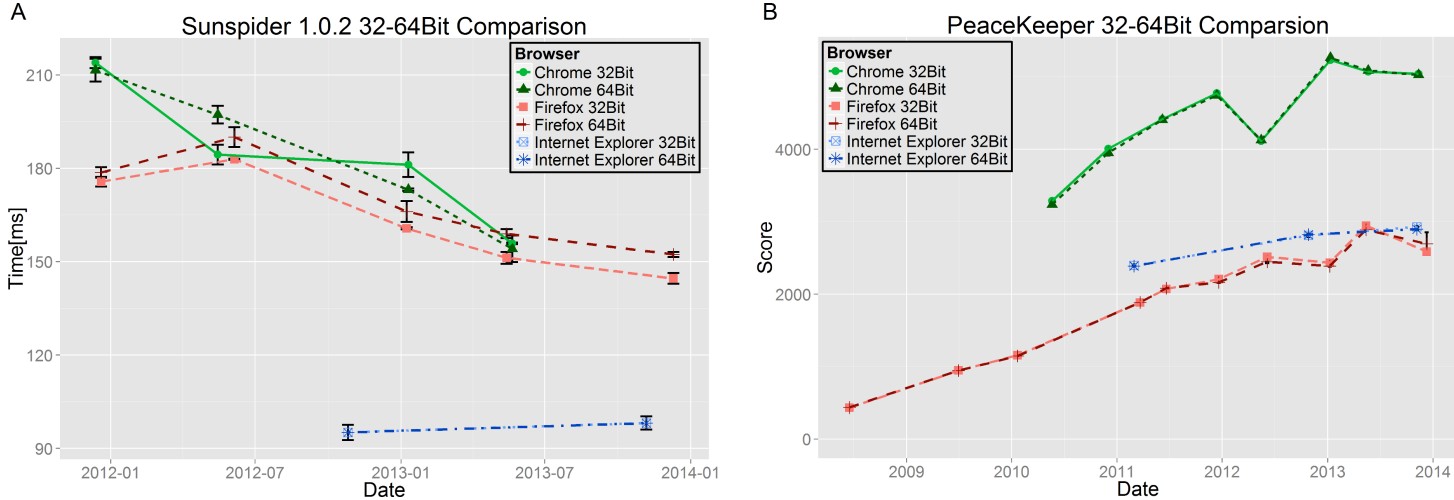

**Figure 3** Comparing the results of two benchmarks on 32 bit and 64 bit platforms.

**Table 2** The benchmarks used in our study, how many repetitions were performed, and which browser versions could not be measured.

| Type | Benchmark | Repetitions | Versions with no data | | |
|---|---|---|---|---|---|
| | | | Internet Explorer | Firefox | Chrome |
| Performance | SunSpider 0.9.1 | 3 | 10,11 | ≥6 | ≥13 |
| | SunSpider 1.0.2 | 3 | 8,9 | ≤5 | ≤12,30,31 |
| | BrowserMark 2.0 | 3 | 8 | 3,3.5,3.6 | 1 |
| | CanvasMark 2013 | 3 | 8 | 3 | 1,2,3 |
| | PeaceKeeper | 3 | | | 1,2,3,4 |
| | Start-up times | 20 | | | |
| Conformance | HTML5 Compliance | N/A | | 4 | 2,7 |
| | CSS3 Test | N/A | | 3 | |
| | Browserscope Security | N/A | | | |

On all performance benchmarks we ran 3 repetitions of each measurement, while for the start-up times measurements we ran 20 repetitions. Error bars are used in the graphs to show the standard error. In all cases, the benchmarks and tests were the only thing that ran on the test machines. The test machines had an Internet connection outside our firewall, so they were not on the local departmental network. This was done for security reasons, as our systems group refused to allow the use of old (and probably vulnerable) browsers within the firewall.

Not all the measurements ran properly with all the versions, especially with earlier versions. The problems were due to the fact that most of the benchmarks were designed and written later than some of the browser early versions, and used some features or technology that were not yet implemented in those early versions. The details are given in Table 2.

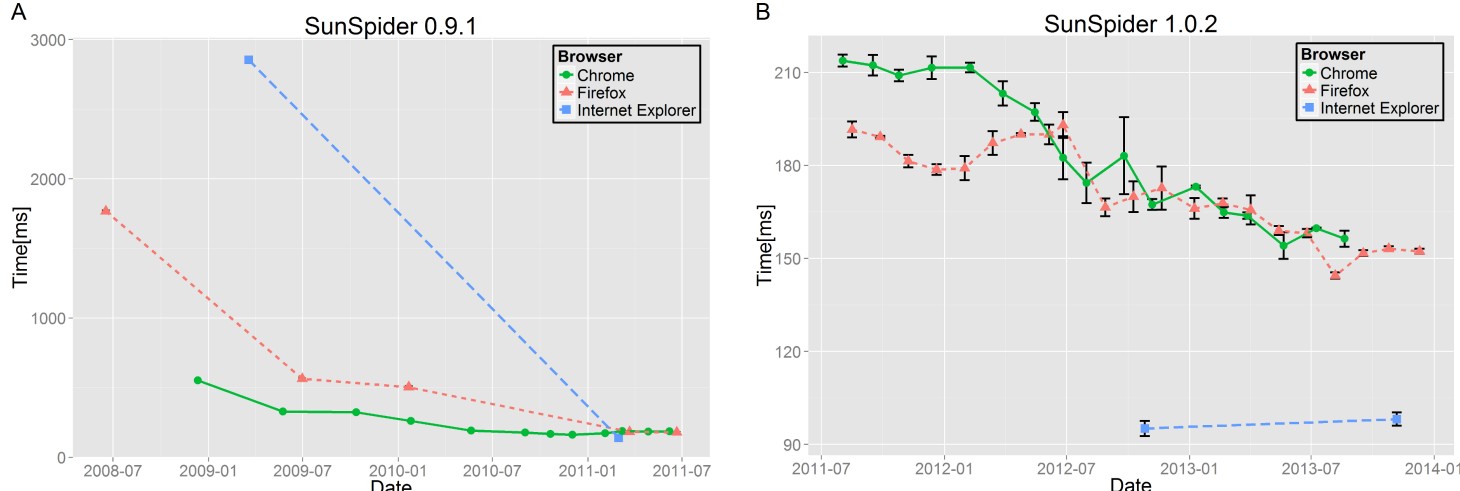

**Figure 4** **SunSpider 0.9.1 and 1.0.2 results.** Note difference in scale for the two versions.

## Performance benchmarks results and interpretation

### *SunSpider*

SunSpider is a well known benchmark developed by WebKit, an open source web browser engine project. Its goal is to measure core JavaScript performance and enable the comparison of different browsers or successive version of the same browser. WebKit designed this benchmark to focus on real problems that developers solve with JavaScript (https://www.webkit.org/perf/sunspider/versions.html). Therefore the benchmark does not include microbenchmarks to evaluate specific language features, but rather tasks such as generating a tagcloud from JSON input, 3D raytracing, cryptography, and code decompression. Moreover, each of the tests is executed multiple times to ensure statistical validity. However, perhaps due to these repetitions, the behavior of the benchmark may actually not mimic real JavaScript work on production sites (*Richards et al., 2011*).

The benchmark measures the time to perform a set of tasks, so lower values are better. In the study we chose to use version 1.0.2, which is the current version and was introduced by WebKit in order to make the tests more reliable (https://www.webkit.org/blog/2364/announcing-sunspider-1-0/, https://www.webkit.org/perf/sunspider-1.0.2/sunspider-1.0.2/driver.html). However, version 1.0.2 didn't work on old browser versions (Table 2). Therefore, we used version 0.9.1 on old browser versions (https://www.webkit.org/perf/sunspider-0.9.1/sunspider-0.9.1/driver.html), specifically those that were tested on the 32 bit machine.

Using SunSpider 0.9.1 we find that when Chrome was introduced it scored significantly better than Internet Explorer and Firefox. In the second version tested of Firefox (Firefox 3.5) the score was greatly improved but still lagged the parallel Chrome version. Although Internet Explorer 8 was released a couple of months after Chrome 1 it was five times slower. It took more than two years for Firefox and Internet Explorer to catch up with Chrome's parallel version (Fig. 4A). In fact, Internet Explorer 9 not only caught up with Chrome but surpassed it. This superior performance has been attributed to its JavaScript

optimization for dead code elimination, which some say was specifically done to boost SunSpider performance (http://blogs.msdn.com/b/ie/archive/2010/11/17/html5-and-real-world-site-performance-seventh-ie9-platform-preview-available-for-developers.aspx, http://digitizor.com/2010/11/17/internet-explorer-9-caught-cheating-in-sunspider-benchmark/).

In the SunSpider 1.0.2 tests Internet Explorer continued to show significantly better results compared to its rivals. Firefox and Chrome showed similar results most of the time (Fig. 4B). For some reasons Chrome versions 30 and 31 had problems with this benchmark, but these were fixed in Chrome 32.

### BrowserMark 2.0

BrowserMark 2.0 is a general browser benchmark developed by Rightware (Basemark), a purveyor of benchmarking and evaluation technology for the embedded systems industry. Originally designed to test mobile and embedded devices, it is nevertheless commonly used to also test desktop browsers. The benchmark tests general browser performance including aspects such as page loading, page resizing, standards conformance, and network speed, as well as WebGL, Canvas, HTML5, and CSS3/3D. The calculated score combines all of these and higher scores are better.

The early versions of Internet Explorer and Firefox did not work with this benchmark (which is understandable given that the benchmark version we used was released only in November 2012). All of the browsers tested showed a distinct improvement trend as new versions were released (Fig. 5). Chrome in all of its versions was better than the equivalent rivals and showed a steady improvement over time. Internet Explorer also showed an improvement over time but always came in last from all the browsers tested. Firefox performance was between Chrome and Internet Explorer. Interestingly, it showed an inconsistent behavior, with the general improvement in benchmark score mixed with local decreases in score.

### CanvasMark 2013

CanvasMark 2013 is a benchmark for performance testing the HTML5 <canvas> tag (http://www.kevs3d.co.uk/dev/canvasmark/). This tag is a container for graphics, which are typically drawn using JavaScript. The benchmark is composed of several stress tests, using elements that are commonly used in games such as operations on bitmaps, canvas drawing, alpha blending, polygon fills, shadows, and drawing text. Each test starts with a simple scene and adds elements until the browser is reduced to a rendering rate of 30 frames-per-second (the rate decreases as the scene becomes more complex). The score is a weighted average of the time the browser managed to perform at above 30 frames-per-second. Higher scores are better.

In this benchmark's documentation there was a note for Chrome users using Windows, encouraging them to change a setting in order to get better results due to a bug in the GPU VSync option for the Windows version of Chrome. However, we did not change the setting since we want to test the versions as the average user would.
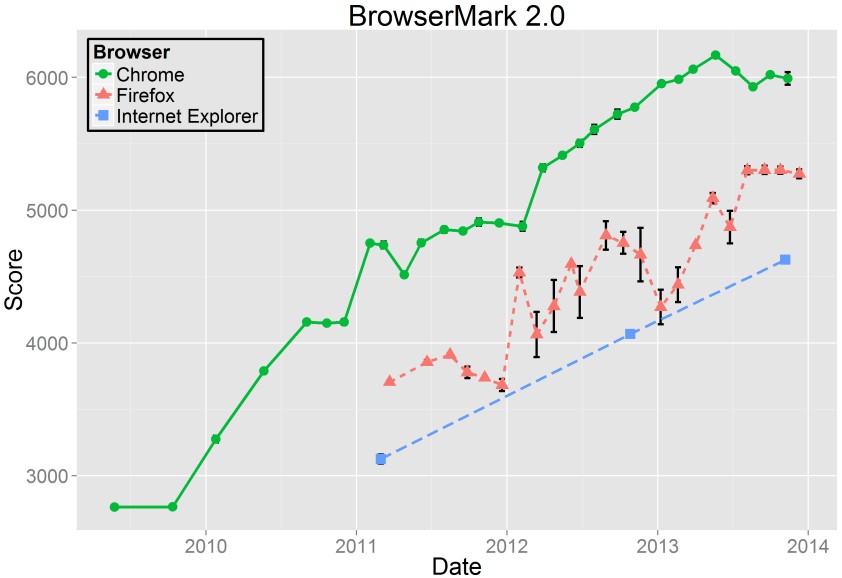

**Figure 5** BrowserMark 2.0 results.

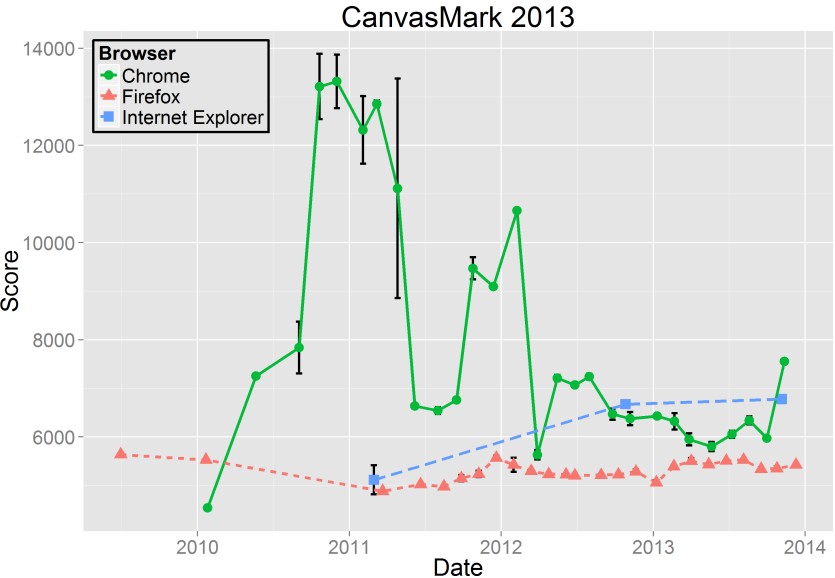

**Figure 6** CanvasMark 2013 results.

The results of running the benchmark show that Chrome exhibited inconsistent results over time (Fig. 6). A great improvement was achieved from version 4 to 7 (version 4 is the first shown, because the benchmark did not run on version 1–3). In contrast there was a sharp decline from version 10 to 12. Later, an improvement occurred from version 14 to 17, immediately followed by a sharp decline of 50% of the score in version 18. But in spite of all these inconsistencies it was still better than Firefox and Internet Explorer during this time. Internet Explorer showed an improvement from version 9 to version 10, when it became the best-performing of the three browsers, due to a deterioration in Chrome's

Tamary and Feitelson (2015), *PeerJ Comput. Sci.*, DOI 10.7717/peerj-cs.28    **11/31**

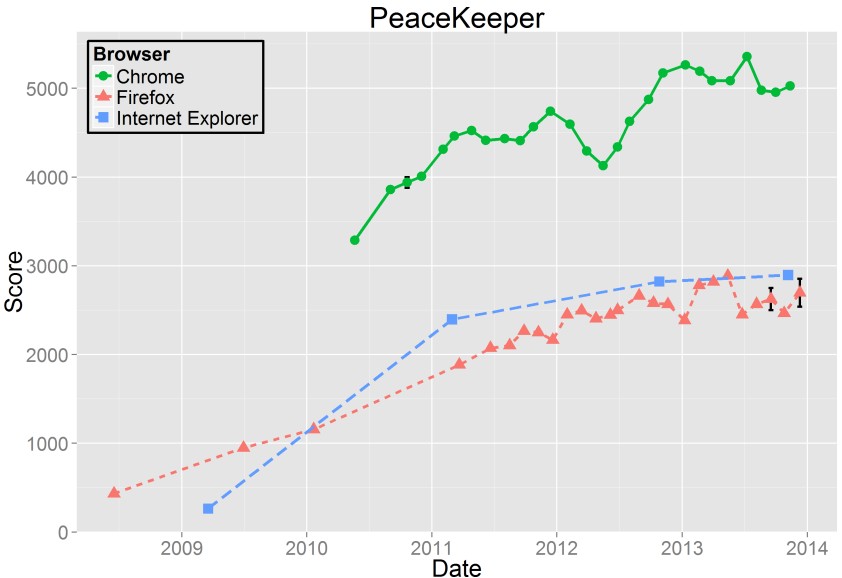

**Figure 7  PeaceKeeper results.**

scores. Chrome surpassed Internet Explorer again only in the last version tested. Firefox had the lowest scores, and does not show any improvement over time.

### PeaceKeeper

PeaceKeeper is a browser benchmark developed by FutureMark, a purveyor of mostly hardware benchmarks for desktop and mobile platforms (http://peacekeeper.futuremark. com/faq.action). (Rightware, the company that developed BrowserMark, was a spinoff from FutureMark.) It includes various tests designed to measure the browser's JavaScript performance, but given that JavaScript is so widely used in dynamic web pages, it can actually be considered to be a general benchmark for browser performance. The tests include various aspects of using JavaScript on modern browsers, such as the <canvas> tag, manipulating large data sets, operations on the DOM tree (the Document Object Model, which describes the structure of a web page), and parsing text. The score reflects processing rate (operations per second or frames per second rendered), so higher is better. In addition the benchmark includes various HTML5 capability checks, such as WebGL graphics, being able to play various video formats, and multithreading support.

Chrome scored noticeably better results compared to its rivals for this benchmark, throughout the period of time that we checked (Fig. 7). However, note that PeaceKeeper did not run on early versions of Chrome (Table 2). Also, while there was a general trend of improvement, it was not monotonic. Firefox and Internet Explorer scored similar results, both showing an improvement over time but still lagging behind Chrome.

## Start-up time measurement methodology and results

An important feature of all browsers, which may affect user satisfaction, is their startup times when they are launched. As we did not find a suitable benchmark that evaluates startup times we conducted specialized measurements to test the browser's cold start-up

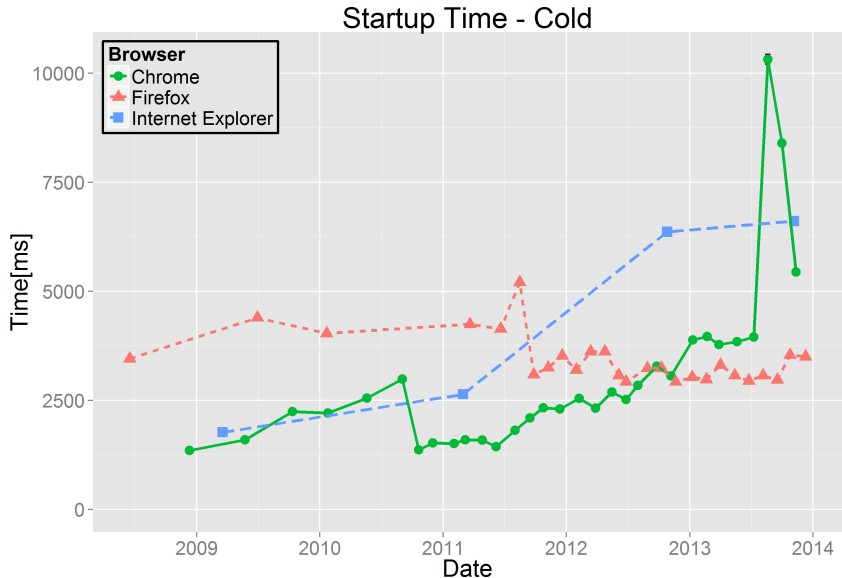

**Figure 8** Cold start-up times.

times. A cold start-up time is when the browser starts for the first time since the operating system was booted.

We tested the start-up times as follows. We wrote a script that runs during the operating system start-up. This script launches the browser one minute after the script starts running. The lag is meant to let the operating system finish loading. A time stamp is created just before launching the browser in order to mark the start time. The browser was set to open with a specially crafted page when it came up. The script passed the time stamp to the crafted page via a URL parameter. The crafted page creates a second time stamp indicating the start of the page processing. The difference between the two time stamps was defined as the browser start-up time. The start-up times are then sent to a server for logging. Advantages of this procedure are, first, that it is independent of network conditions, and second, the test is similar to the user's real experience of launching the browser and loading the first page.

The first versions of Chrome were the fastest to load (Fig. 8). However, as Chrome's development advanced, its start-up times crawled up. In Chrome version 7 the start-up times improved dramatically, but then continued to crawl up from version 13. In version 29 there was a spike in the start-up time, a 2.5 fold increase compared to the previous version, followed by a partial correction in versions 30 and 31. Surprisingly Firefox start-up times look steady with a slight decrease, notably in version 7. As a result, while it was the slowest by a wide margin in 2009, it became the fastest in 2013. Internet Explorer start-up time were initially similar to those of Chrome, but then increased consistently over time, making it roughly twice as slow as the others in recent years (except for the spike in Chrome performance since version 29).

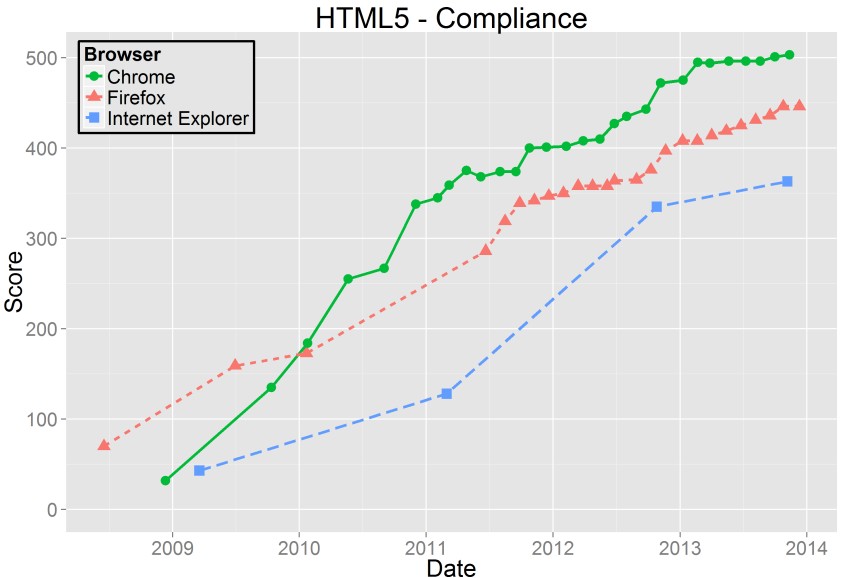

**Figure 9** HTML5 Compliance results.

## Conformance benchmarks results and interpretation

### HTML5 Compliance

HTML (Hyper-Text Markup Language) is the language used to describe web pages, and the current version is HTML5. HTML5 introduced features like the <canvas> tag for use by multimedia applications, and integrated SVG (Scalable Vector Graphics) and MathML (for mathematical formulas). The first working draft of HTML5 was published in 2008, and the standard was finally approved in 2014, so its definition process fully overlaps the period of Chrome's rise.

The HTML5 Compliance benchmark consists of three parts. The main part is checking the conformance of the browser to the HTML5 official specification. The second part is checking specifications related to HTML5 such as WebGL. The third part is checking the specification for experimental features that are an extension to HTML5 (http://html5test.com/about.html). The score is the sum of points awarded for each feature that is supported.

The results for this benchmark show that all the browsers improve over time. Firefox had the best score until version 3.6, and after that Chrome version 4 and up had the best score (Fig. 9). Internet Explorer always had the lowest score.

### CSS3 test

CSS (Cascading Style Sheets) is the language used to describe the style of HTML pages. For example, using CSS one can set the style for web page headings and make it different from the default of the browser. The current version is 3, although level-4 modules are being introduced.

CSS3 Test checks how many CSS3 elements in the W3C specification does a certain browser recognize (http://css3test.com/). This means CSS3 Test only checks the recogni-

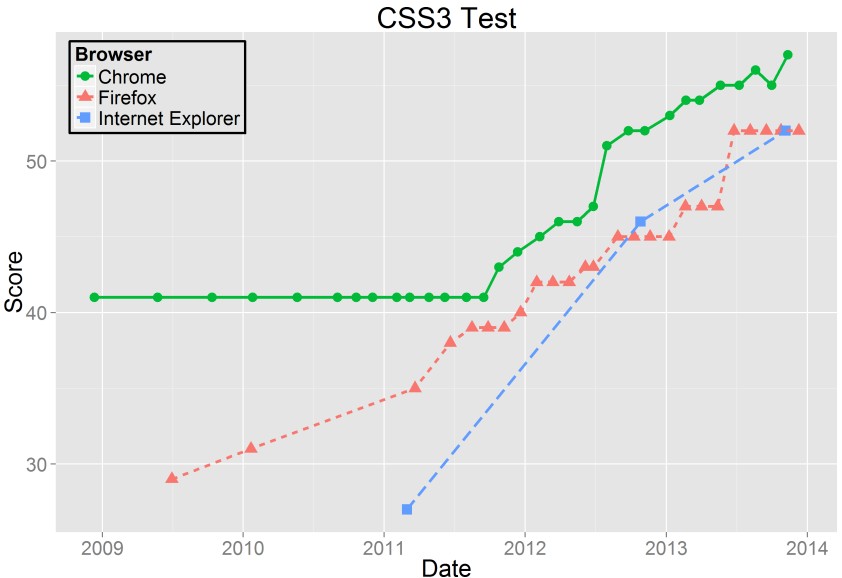

**Figure 10 CSS3 Test results.**

tion itself but does not check the implementation or the quality of the implementation, namely whether the resulting rendition of the web page indeed looks like it should.

Interestingly Chrome's score did not change in the first three years, though it still managed to have a better score than its rivals. From version 15 Chrome consistently improved until the last version tested, remaining better than its rivals all along (Fig. 10). Firefox showed several improvements in a stepwise manner. Internet Explorer had the lowest score in the first version tested (version 9) but improved its score dramatically in versions 10 and 11, achieving essentially the same level as Firefox.

### Browserscope security

Browserscope is a community-driven project which profiles various aspects of web browsers. One of these is the obviously important feature of security. Specifically, Browserscope Security is a collection of tests meant to check "whether the browser supports JavaScript APIs that allow safe interactions between sites, and whether it follows industry best practices for blocking harmful interactions between sites" (http://www. browserscope.org/security/about). For example, one of the tests checks whether the browser has native support for JSON parsing, which is safer than using eval. The score is simply how many tests passed. While this is not strictly a conformance test, as there is no official standard, we include it due to the importance of security features on the Internet.

The results are that all three browsers exhibited a general (although not always monotonic) improvement in their security results over time. The relative ranking according to these tests is very consistent between browser versions (Fig. 11). Across practically the whole period Chrome had the highest score, Firefox had the lowest, and Internet Explorer was in between. The only exception is a large dip in score for Chrome versions 2 and 3, where version 2 was the worst of all parallel browser versions. This was

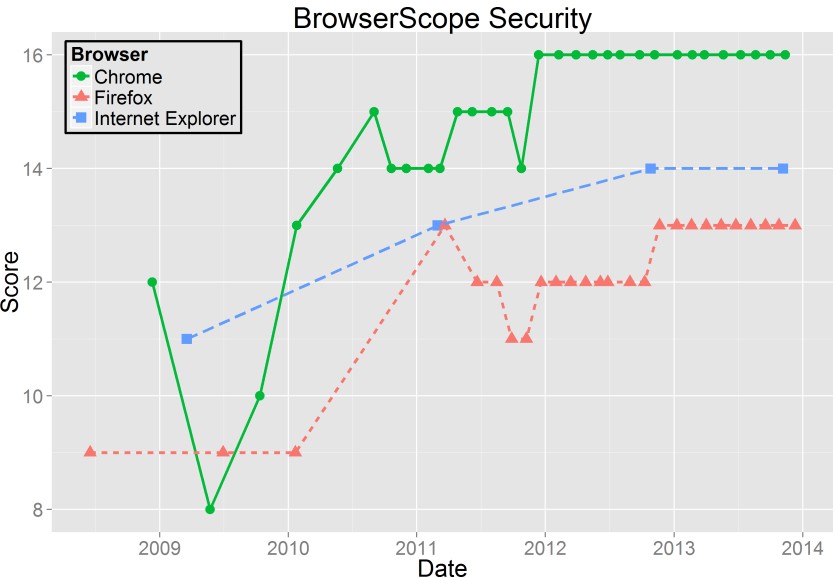

**Figure 11  Browserscope Security results.**

surprising because of the overall consistency, and the fact that in the first version released Chrome had the highest score compared to its rivals.

## Additional results with Opera

Our main measurements focused on the three top browsers, which together control more than 90% of the desktop market. But when considering the relative importance of technical issues as opposed to marketing, we felt the need to also consider the smaller browsers. This is especially important in the early years, when Chrome's market share was low, and the question was what enabled Chrome to surge ahead while other browsers were left behind.

We therefore conducted a few additional measurements using Opera. We focused on Opera and not on Safari for two reasons. First, Opera has a reputation for being an innovative and technologically advanced browser. Second, Safari is specifically targeted for Apple platforms, and therefore is not really part of the same desktop market as the other browsers we are studying.

Not all versions of Opera were tested, as many of the benchmarks did not run properly on early versions. The results were that Opera performance was generally inferior to that of Chrome (two examples are shown in Fig. 12). In some benchmarks, notably BrowserMark and Browserscope Security, its scores were actually lower than for all other browsers for many years. The sharp improvement in BrowserMark shown in Fig. 12A is probably due to the move to using WebKit (and thus the same rendering engine as Chrome) in version 15 (https://dev.opera.com/blog/300-million-users-and-move-to-webkit/); similar improvements were also seen in some other benchmarks in this version. In other benchmarks, such as HTML5 Compliance and CSS3 Test, Opera's results were similar to those of Firefox throughout. The only benchmark in which Opera was the best browser for a considerable period was CanvasMark, but this period only started in 2012 (and performance dropped in version 15).

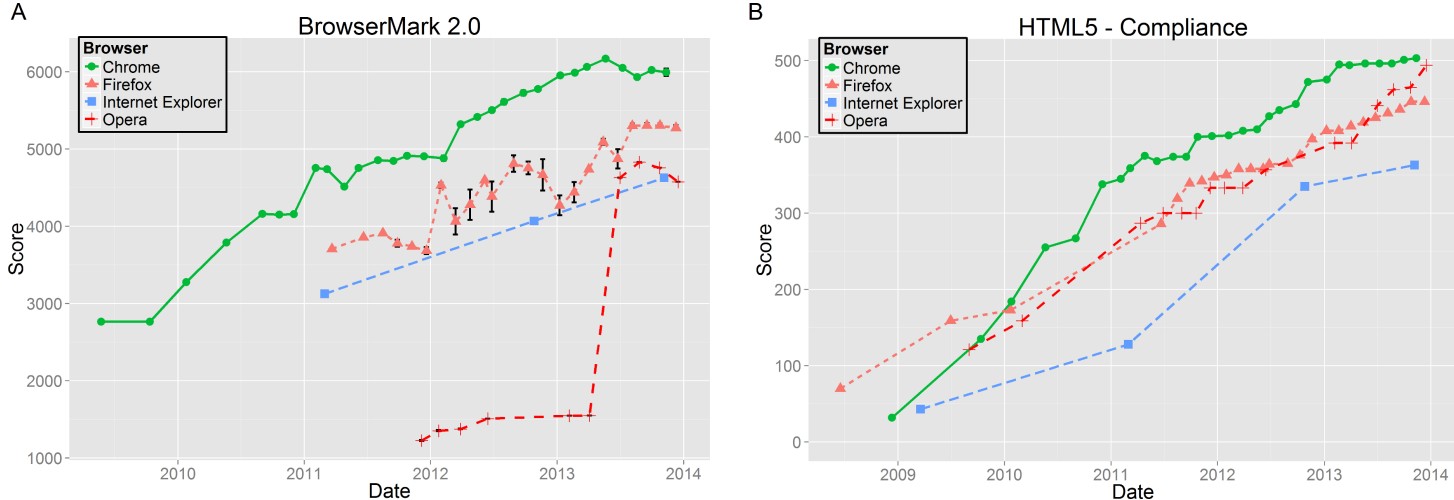

**Figure 12** Sample Opera results overlaid on the previous results.

# FEATURE SELECTION AND RELEASE

Another aspect in which the browsers differ from one another is their feature sets: all obviously have the same basic features allowing users to browse the web and display web pages, but new features are added all the time as web usage continues to evolve. However, not all features have the same importance, so it is advantageous for a browser to have the most meaningful features as early as possible.

In this section we present the methodology and results pertaining to answering research question (2), namely which browsers released features early and which browsers lagged in releasing features. In addition we wanted to evaluate the importance of each feature. We used an online survey to assess the importance of each feature to the end users.

## Experimental design and methodology

The investigation of the features embodied in each browser and their release times involved the following steps:

1. Listing of major features of modern browsers.

2. Establishing the release date of each feature by each browser

3. Identification of features that differentiate between the browsers

4. Conducting an online survey of web users to assess the relative importance of the different features to end users

5. Performing a statistical analysis of the relative importance (according to the survey) of features that each browser released earlier or later than other browsers.

The following subsections provide details regarding these steps.

### *Feature selection*

We identified 43 features which in our opinion represent a modern browser. These features are listed in Tables 3 and 4. The release date of each feature in each browser was ascertained

**Table 3  Features not used in comparisons as they did not reflect differences between browsers.**

| # | Feature | Explanation |
|---|---------|-------------|
| *Pre Chrome 1* | | |
| 1 | Bookmark management | Allows the user to organize/delete/add bookmarks |
| 2 | Password management | Allows the browser to remember credentials to a certain web site upon the user's request |
| 3 | Search engine toolbar | Easy access to a search engine from the browser tool bar |
| 4 | Tabbed browsing | The ability to browse multiple web pages in a single browser window |
| 5 | Pop-up blocking | Blocks pop-ups that the user didn't explicitly ask for |
| 6 | Page zooming | Scale the text of a web page |
| 7 | History manager | Manages history of recent web pages that the user browsed |
| 8 | Phishing protection | Block or warn when surfing to web pages that masquerade as another (legitimate) website |
| 9 | Privacy features | Manages the user preferences regarding passwords, history, and cookie collection |
| 10 | Smart bookmarks | bookmarks that directly give access to functions of web sites, as opposed to filling web forms at the respective web site for accessing these functions |
| 11 | Tabbing navigation | The ability to navigate between focusable elements with the tab key |
| *Released at same time* | | |
| 1 | Access keys | Allows you to navigate quickly through a web page via the keyboard |
| 2 | Adaptive address bar | Suggest webpages as you type an address or search keywords from your history or from a search engine |
| 3 | Full page zoom | Scales the whole page, including images and CSSs |
| 4 | Hardware acceleration | Allows the GPU to help the browser to speed up certain tasks that the GPU is more capable for |
| 5 | Incognito | Allows the user to browse the web with reduced identifiable trace (notably doesn't allow cookies) |
| 6 | Reopen closed tabs | Reopen a recently closed tab |
| 7 | Full screen | Displays the page in full screen mode |

by the combination of reading release documentation and checking whether features exist in the different versions we downloaded.

The third step, namely identifying features that differentiate between the browsers, is based on the release dates. The reason is that features only differentiate between browsers if they are released at different times. As Chrome 1.0 was our starting point, 11 features included in this version which had already been included also by the competing browsers were excluded from consideration, as they did not confer any competitive advantage to any browser in the context of our study. For example, this included multiple tab browsing. Seven further features were excluded because they were released at about the same time by all three browsers, so they too did not confer any competitive advantage (Table 3 and see below). Subsequently, the study was conducted based on the 25 remaining features. These features are listed in Table 4. Note that the features are listed in a random order.

### Feature release margins

As the three browsers are developed by different organizations, the release dates of new versions are of course not coordinated. We therefore faced the challenge of defining what it means for one browser to release a feature ahead of another. We elected to use a conservative metric for this concept.

We had already dated the release of each of the 25 selected features in each browser (Table 5). We then developed a metric which states whether a certain browser released a

**Table 4  Selected features used in comparisons (arbitrary order).**

| # | Feature | Explanation |
|---|---------|-------------|
| 1 | Add-ons manager | Allows you to disable/remove previously installed add-ons |
| 2 | Download manager | Allows you to view/pause current downloads and view previously downloaded files |
| 3 | Auto-updater | Silently & automatically updates the browser if there's a new version |
| 4 | Caret navigation | Allows you to navigate through a site using the arrow keys (just like in any document processor e.g., in Microsoft Word) |
| 5 | Pinned sites | Allows you to have faster access to your favorite sites like Facebook or your Email provider |
| 6 | Sync | Allows you to sync you favorites/preferences/saved passwords etc. through computers and platforms |
| 7 | Session restore (automatically) | Upon a crash, the browser will restore the sites you were surfing before the crash |
| 8 | Crash/security protection | Allows you to continue browsing although a site/plugin crashed or hanged |
| 9 | Malware protection | Enables the browser to warn and block suspicious sites that are known to be harmful |
| 10 | Outdated plugin detection | Allows the browser to detect if a plugin has become incompatible/vulnerable with the browser's version |
| 11 | Do not track | Allows the browser to request sites not to track the browsing |
| 12 | Themes | Allows you to personalize the browser appearance by changing the skin |
| 13 | Experimental features | Allows you to try experimental features in the browsers that aren't turned on by default |
| 14 | Multiple users | Allows you to have multiple profiles (different bookmarks/saved passwords/history) on the same computer user |
| 15 | Apps | Allows you to install applications that will run in the browser (like games or other applications) |
| 16 | Developer tools | Allows you to examine a site's interaction with the browser |
| 17 | Personalized new tab | Allows you to see your most visited sites upon the launch of the browser (the first tab that is opened on launch of the browser) |
| 18 | Click-to-play | Disables the automatic launch of a plugin's content. The user must explicitly click on the flash/applet in-order to load and play it |
| 19 | Print preview | Allows the user to view the page before printing it |
| 20 | Per-site security configuration | Allows you to control which sites will block popups/cookies/images/scripts/plug-ins etc. |
| 21 | Web translation | Allows the browser to translate a page automatically to a desired language |
| 22 | Spell checking | Marks misspelled input you typed and corrects it |
| 23 | Built-in PDF viewer | Allows the browser to open PDF files without any 3rd party plugins |
| 24 | Sandboxing | A security concept that certain parts of the browser run individually with restricted privileges only |
| 25 | RSS reader | Allows the browser to know when a certain site, that supports RSS, was updated. News feeds etc. |

feature ahead of a competitor by "a meaningful margin" and/or whether a certain browser lagged a competitor by "a meaningful margin". A browser was awarded a "win" if it released a feature ahead of all its competitors, and a penalty or "loss" was given if a browser lagged all its competitors or did not released a certain feature at all. Note that each feature can have a maximum of one "winner" and a maximum of one "loser". If a feature had neither a "winner" nor a "loser" it was excluded from the study as no browser had a competitive advantage or disadvantage.

"A meaningful margin" was defined as more than one release cycle, that is, when it took the competitors more than one version to include the feature after it was initially introduced. For example, "personalized new tab" was introduced in Chrome 1. At the time the most recent versions of Internet Explorer and Firefox were 7 and 3, respectively. The feature was subsequently released in Internet Explorer 9 and Firefox 13, meaning that this was a meaningful margin. Had the feature been released in Internet Explorer 8 or Firefox

**Table 5  Feature release versions and survey results.** W and L denote wins and losses, respectively.

| # | Feature | Feature release version | | | Survey results | | | | |
|---|---------|------------------|---------|--------|---|---|---|---|---|
| | | **Explorer** | **Firefox** | **Chrome** | **1** | **2** | **3** | **4** | **5** |
| 1 | Add-ons manager | pre 8 | pre 3 | 4 L | 10 | 24 | 39 | 100 | 81 |
| 2 | Download manager | 9 L | pre 3 | 1 | 7 | 24 | 48 | 84 | 91 |
| 3 | Auto-updater | 9 | 16 L | 1 W | 28 | 39 | 66 | 62 | 59 |
| 4 | Caret navigation | 8 | pre 3 | none L | 56 | 61 | 49 | 43 | 45 |
| 5 | Pinned sites | 9 | 5 L | 2 | 45 | 36 | 47 | 60 | 66 |
| 6 | Sync | 10 L | 4 | 4 | 56 | 33 | 47 | 62 | 56 |
| 7 | Session restore (automatically) | 10 L | pre 3 | 1 | 16 | 28 | 28 | 62 | 120 |
| 8 | Crash/security protection | 8 | 20 L | 1 | 6 | 13 | 37 | 110 | 88 |
| 9 | Malware protection | 9 L | 3 | 1 | 9 | 13 | 41 | 73 | 118 |
| 10 | Outdated plugin detection | none L | 3 W | 10 | 15 | 56 | 74 | 77 | 32 |
| 11 | Do not track | 9 | 4 | 23 L | 21 | 28 | 43 | 85 | 77 |
| 12 | Themes | none L | pre 3 W | 3 | 134 | 60 | 35 | 21 | 4 |
| 13 | Experimental features | none L | pre 3 W | 8 | 72 | 76 | 51 | 43 | 12 |
| 14 | Multiple users | none L | pre 3 W | 12 | 110 | 50 | 40 | 37 | 17 |
| 15 | Apps | none L | 16 | 9 W | 77 | 63 | 51 | 47 | 16 |
| 16 | Developer tools | 8 | 4 L | 1 | 51 | 40 | 44 | 51 | 68 |
| 17 | Personalized new tab | 9 | 13 L | 1 W | 48 | 61 | 67 | 49 | 29 |
| 18 | Click-to-play | none L | 8 | 8 W | 31 | 54 | 75 | 53 | 41 |
| 19 | Print preview | pre 8 | pre 3 | 9 L | 19 | 21 | 58 | 92 | 64 |
| 20 | Per-site security configuration | 11 L | 3 W | 5 | 11 | 33 | 87 | 75 | 48 |
| 21 | Web translation | none | none | 5 W | 50 | 66 | 59 | 55 | 24 |
| 22 | Spell checking | 10 L | pre 3 | 1 | 20 | 35 | 44 | 78 | 77 |
| 23 | Built-in PDF viewer | none L | 19 | 8 W | 11 | 28 | 42 | 80 | 93 |
| 24 | Sandboxing | pre 8 | none L | 1 | 54 | 43 | 59 | 51 | 47 |
| 25 | RSS reader | 8 | pre 3 | none L | 119 | 65 | 34 | 22 | 14 |

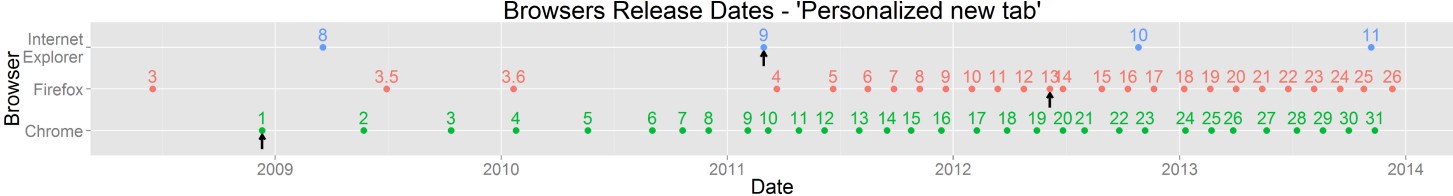

**Figure 13  Feature release margin example.** The "personalized new tab" feature was released in Chrome 1, Internet Explorer 9, and Firefox 13 (marked with arrows)

3.5 it would not have counted as a meaningful margin, despite being later than Chrome 1. Furthermore, Firefox lagged Internet Explorer in the release of the feature in a meaningful margin (Fig. 13). So in this case Chrome received a "win" and Firefox received a "loss". All the release versions and their identification as wins or losses are shown in the results in Table 5.

Note that the definition of the release margin is based on releases of new versions, and not on absolute time. This definition gives an advantage to browsers that are released

infrequently. For example, any innovations included in Chrome versions 2 to 9—a span of nearly two years—and included in Internet Explorer 9 would not be considered to have a significant margin, because Microsoft did not release any versions during all this time. Consequently our results may be considered to be conservative.

### *Feature importance survey*

To assess the relative importance of the 25 different features, we created an online survey that lists and explains these features. Survey participants were asked to evaluate the importance of each feature relative to other listed features on a discrete scale of 1 (least important) through 5 (most important). The features were listed in the same random order as in Table 4.

The intended audience was people who spend many hours a day on the World Wide Web. The survey was published on Reddit (sub-reddit /r/SampleSize) (http://www.reddit.com/r/SampleSize/) and on CS Facebook groups of the Hebrew University and Tel-Aviv University in Israel. 254 people answered the survey, and the distribution of results is shown in Table 5. The statistical analysis was performed on all of the participants.

## Statistical analysis procedure

Opinion surveys like the one we conducted are commonly analyzed by calculating the average score received by each entry, and considering these to be the averages of samples of different sizes and unknown variance. Then a test such as Welch's *t*-test is used to check whether or not these averages are significantly different. However, such an approach suffers from a threat to construct validity, because averaging implicitly assumes that the scale is a proper interval scale, meaning that the difference between 1 and 2 is the same as between 2 and 3, 3 and 4, and 4 and 5. But given that these numbers represent subjective levels of importance, this is not necessarily the case. Moreover, different people may use the scale differently. Therefore both the averaging and the statistical test are compromised.

Another problem with human users is that some of them are hard to please, and always use only the bottom part of the scale, while others are easy to please, and focus on the top part of the scale. To reduce this danger our survey participation instructions included the following:

> "For every feature please choose how important this particular feature is compared to other features in the survey. Please try to use the full scale from 'least important' to 'most important' for the different features. You can change your marks as often as you wish before submitting."

And indeed, checking our data we found that most respondents actually used the full scale from 1 to 5, with an average near 3. These findings imply that we do not need to perform adjustments to the data to compensate for potentially different behaviors (*Rossi, Gilula & Allenby, 2001*).

Nevertheless, comparing average scores is still not justifiable. We therefore use an analysis method due to *Yakir & Gilula (1998)*, where brand *A* is judged to be superior to brand *B* if the distribution of opinions about *A* dominates the distribution of opinions

about *B* in the stochastic order sense. Note that in our case the "brands" are not Microsoft, Google, and Mozilla, but rather the sets of features which represent the "wins" or "losses" of each browser. This will be clear in the results of 'Results'.

Mathematically stochastic order is expressed as $\forall s : F_A(s) \leq F_B(s)$, where $F_A$ and $F_B$ are the cumulative distribution functions of the opinions regarding *A* and *B*, respectively. Graphically, the plot of $F_A$ is lower and to the right of the plot of $F_B$, and it accumulates more slowly. In simple terms this means that for each level of opinion 1 to 5 the probability that *A* receives a score of at least this level is higher than the probability that *B* receives such a score. However, in many cases one distribution does not dominate the other (and their graphs cross each other). It is then necessary to adjust the data by grouping brands and/or score levels together until dominance is achieved (*Gilula, 1986*; *Gilula, Krieger & Ritov, 1988*; *Gilula & Krieger, 1989*; *Ritov & Gilula, 1993*).

In more detail, the analysis procedure is as follows (*Yakir & Gilula, 1998*):

1. *Identify subsets of homogeneous brands*. Ideally, for all pairs of brands, the distribution of scores of one brand will dominate the distribution of scores of the other brand. This will induce a full order on the brands. But in reality there may be certain subsets of brands that are incomparable, and do not dominate each other. These subsets need to be identified, and the ranking will then be between subsets instead of between individual brands.

   The subsets are found by an agglomerative single-link clustering algorithm. Initially all pairs of individual brands are compared, and the chi-squared statistics computed. If the minimum statistic value obtained is below a predefined critical value, the two distributions are considered the same and the brands are combined into a joint subset. In subsequent steps, when subsets are being considered, the maximal statistic among all pairs (where one brand comes from the first subset and the other brand from the second subset) is compared to the critical value.

   The suggested critical value is the upper $\alpha$ percentile from the chi-square distribution with $J - 1$ degrees of freedom, where $J$ is the number of score levels in the distribution (in our case, 5) and $\alpha = 0.1$ (or another value chosen by the analyst).

   A chi-square-based test is then applied to test whether the obtained partitioning is significant, as described in *Yakir & Gilula (1998)*. The result of this step is then one or more subsets of brands, which are heterogeneous relative to each other, but the brands within each subset are homogeneous.

2. *Find the widest collapsed scale*. Even when brands (or subsets of brands) have heterogeneous distributions of scores, the distributions may not dominate each other in the stochastic order sense. This happens if the distributions cross each other. However, it is always possible to create a dominance relationship by collapsing adjacent scores and thereby reducing the fidelity of the distributions.

   The problem is that collapsing can be done in many different ways, and the selected collapsing may affect the resulting dominance order. We therefore need to define which collapsing is better. The suggested approach is to strive for minimal loss of information,

in the sense of preserving as many of the original scores as possible. Hence we are looking for the widest collapsed scale that nevertheless leads to dominance.

Technically, the procedure is as follows. Given all the subsets of brands, we consider all possible orders of these subsets. For each such order we find the collapsing that leads to dominance in this order (if such a collapsing exists). The order that is supported by the widest collapsing is then selected. This implies using the collapsing which retains the highest number of distinct scores.

3. *Note the stochastic order between the subsets of brands.* At this point a well-defined stochastic order is guaranteed to exist. This order is the result of the analysis.

4. *Verify statistical significance.* Collapsing score levels leads to loss of information relative to the original data. A chi-square-based test is used to demonstrate that the loss is not significant, and therefore the results will still reflect the original data. For details see *Yakir & Gilula (1998)*.

In our case the brands are the features of the browsers. But we don't really care about ranking the individual features. Rather, we want to rank sets of features. For example, we can take the set of features that were Chrome "wins", and compare it to the set of features that were Chrome "losses". If the first set turns out to be more important to users, then this testifies that Chrome project managers chose wisely and invested their resources in prioritizing the more important features first.

To perform these calculations we used the Insight for R v0.4 software package which implements this approach.[2] Given the adjusted (collapsed) data, we also calculate the polarity index. The polarity index is the ratio of users who considered features important (levels 4 and 5) to the rest (levels 1 to 3). A polarity index less than 1 indicates that the balance is skewed towards not important, while a polarity index higher than 1 indicates that user opinion is skewed towards most important. Unlike average scores, the polarity index has a direct quantitative meaning and therefore the indexes of different brands can be compared to each other.

[2] The software and statistics advice were kindly provided by professor Zvi Gilula.

## Results

### Early release of features

In order to analyze which browser released features earlier than its competitors we identified the "wins" and "losses" of each browser, as indicated in Table 5. Our results show that Chrome received a "win" in 6 features and Firefox in 5 features. In contrast, Internet Explorer did not receive any "wins", and 14 features did not have a "winner". Chrome received a "loss" in 5 features, Firefox in 6 features, and Internet Explorer in 13 features. Here only one feature was not ascribed as a "loss" to any of the browsers (the "web translation" feature).

These results already show that Chrome tended to release new features ahead of the other browsers, with Firefox being a very close second. Internet Explorer lagged far behind both of them, as it did not release *any* feature ahead of the competition and it was the last to release half of the features in the study.

**Table 6** Comparison between Chrome and Firefox "wins".

| Rank | Browser | No. of wins | Importance score dist. | | | | | Polarity index | |
|---|---|---|---|---|---|---|---|---|---|
| | | | 1 | 2 | 3 | 4 | 5 | | |
| 1 | Chrome | 6 | 0.16 | 0.20 | 0.24 | 0.23 | 0.17 | 0.67 | |
| 2 | Firefox | 5 | 0.27 | 0.22 | 0.23 | 0.20 | 0.09 | 0.40 | |
| 3 | Internet Explorer | 0 | N/A | N/A | N/A | N/A | N/A | N/A | |

### Importance comparisons

Mere counting of "wins" and "losses" as done above does not indicate whether the features released early by Chrome were indeed the more important ones. We therefore conducted an analysis of importance by comparing the distributions of importance scores given to the sets of features that were "wins" and "losses". Specifically, we performed an analysis of the "wins" of different browsers, an analysis of their "losses", and a specific analysis of the "wins" versus the "losses" of Chrome.

*Wins.* The results of comparing the user opinions regarding the feature sets where each browser "won" is shown in Table 6. A stochastic order of the response levels was present without any adjustments, with Chrome ranked first and Firefox second. Since Internet Explorer did not have any "wins" it was ranked last. The Polarity Index of Chrome and Firefox were 0.67 and 0.40, respectively. While both are smaller than 1, the features in which Chrome received a "win" were still more important to the end user, since the Polarity index was higher. The direct quantitative meaning is that for Chrome users considered the "winning" features to be important $\frac{2}{5}$ of the time, whereas for Firefox they considered them to be important only about $\frac{2}{7}$ of the time.

*Losses.* Given limited resources the developers of a browser cannot do everything at once, so the implementation of select features must be delayed. Under such circumstances it is best to delay those features that will not be missed by many users, namely those that are considered less important. Therefore, a lower ranking and a lower polarity index are favorable when comparing feature sets which are "losses".

The "loss" scores distributions of Firefox and Internet Explorer showed the same trends and could not be distinguished one from the other, so they were clustered together. In order to achieve dominance the ranking algorithm collapsed importance score levels 3 and 4 (Table 7). The result after these adjustments was that Firefox and Internet Explorer were ranked on top and Chrome was ranked lower. This means that the features in which Firefox and Internet Explorer received a "loss" were more important to the end users. However, it should be noted that the differences in the distributions were actually very small, so this difference is most probably meaningless. The Polarity Index could not be calculated in the regular way due to the unification of levels 3 and 4. The results given in the table are therefore the ratio of levels 3 to 5 to levels 1 and 2, making them higher than in other comparisons. They are close to each other, but still Chrome is a bit lower, which is better in this case.

**Table 7** Comparison between Chrome and Firefox/Internet Explorer "losses". Note that is this case being ranked lower is better.

| Rank | Browser | No. of losses | Importance score dist. | | | | Polarity index[*] | |
|---|---|---|---|---|---|---|---|---|
| | | | 1 | 2 | 3–4 | 5 | | |
| 1 | Firefox | 6 | 0.17 | 0.16 | 0.45 | 0.22 | 2.03 | |
| | Internet Explorer | 13 | | | | | | |
| 2 | Chrome | 5 | 0.18 | 0.16 | 0.44 | 0.22 | 1.94 | |

**Notes.**

[*] The polarity index is calculated differently than in other cases because scores 3 and 4 were collapsed.

**Table 8** Comparison between Chrome "wins" and "losses".

| Rank | Class | No. of features | Importance score dist. | | | | Polarity index | |
|---|---|---|---|---|---|---|---|---|
| | | | 1–2 | 3 | 4 | 5 | | |
| 1 | Losses | 5 | 0.33 | 0.18 | 0.27 | 0.22 | 0.96 | |
| 2 | Wins | 6 | 0.36 | 0.24 | 0.23 | 0.17 | 0.67 | |

*Chrome wins and losses.* Finally, we compared the features that Chrome "won" with those that it "lost". In order to achieve a stochastic order the algorithm collapsed levels 1 and 2 together. Interestingly the "losses" won, meaning that they were considered more important (Table 8). The Polarity Index of the "wins" and the "losses" were 0.67 and 0.96, respectively, meaning the features which Chrome released ahead of its rivals were considered important to the users about 40% of the time, whereas those in which it lagged behind were considered important nearly 50% of the time. Thus the prioritization used in developing Chrome was better than that of its rivals (as shown in the two previous analyses), but it was far from perfect.

# DISCUSSION

## Summary of results

We tested the performance of the three dominant browsers, Chrome, Firefox, and Internet Explorer, and to a lesser degree also the Opera browser, using a wide set of commonly used benchmarks and across a long period of time. The results, presented in 'Performance benchmarks results and interpretation' through 'Additional results with opera' and summarized in Table 9, show that Chrome generally had an advantage over its competitors, both in terms of performance and in terms of conformance with standards.

More specifically, Chrome achieved better results throughout in five of the tests: BrowserMark 2.0, PeaceKeeper, HTML5 Compliance, CSS3 Test, and Browserscope Security. Firefox achieved better results only in the start-up times test, and that only towards the end of the study period. Interestingly, Chrome start-up times results may indicate that Chrome suffers from a feature creep impacting its start-up times. Internet Explorer achieved better results only in SunSpider, in the second half of the study

**Table 9  Summary of benchmark results.**

| Benchmark | Result |
|---|---|
| SunSpider | Chrome was best through 2010, now Internet Explorer is significantly better |
| BrowserMark 2.0 | Chrome is best, Explorer worst |
| CanvasMark 2013 | Chrome is relatively good but inconsistent, Firefox worst |
| PeaceKeeper | Chrome is significantly better |
| Start-up times | initially Chrome was better but now Firefox is better, Explorer has deteriorated |
| HTML5 Compliance | Chrome is better, Explorer worst |
| CSS3 Test | Chrome is better |
| Browserscope Security | Chrome is better, Firefox worst |

period. Moreover, Chrome was not worse than both competing browsers in any of the benchmarks, while Firefox and Internet Explorer were each the worst browser in two cases.

In addition we compared the release dates and importance of 25 specific features, as described in 'Results'. Eleven features had a "winner", meaning that they were released by one browser ahead of the others by a meaningful margin. All but one also had a "loser", that is a browser that lagged behind by a significant margin. The relatively low fraction of features that had a "winner" (and the fact that 7 features were excluded from the study because they did not have a "winner" nor a "loser") indicates that the development of each browser is not isolated from its rivals. As a result, some features are released at about the same time by two or even all three browsers. On the other hand, some browsers still managed to release a fair number of innovative features: Chrome and Firefox received 6 and 5 "wins", respectively. Internet Explorer on the other hand did not receive any "wins" and had the most "losses", 13. Chrome and Firefox had 5 and 6 "losses", respectively.

Although Chrome and Firefox received similar numbers of "wins" the feature importance survey showed that features in which Chrome "won" were more important to the users than features in which Firefox "won". Likewise, features in which Chrome "lost" were less important to users than the features in which Firefox and Internet Explorer had "lost", but in the case of losses the difference was marginal. Interestingly, Chrome "losses" were actually more important to users than its "wins".

Ideally a browser should release the most important features to users first, and in case it has to lag in the release of certain features they should be of less importance to users. The results indicate that Chrome project managers were somewhat better at releasing important features first than the project managers of competing browsers. This means that they generally made better choices than their rivals. However, they did not manage to focus on only the important features, and when they lagged in feature release, these features were sometimes actually more important to users.

## Implications for software development

While not the focus of our study, our results can be used to glean some insights into basic questions in large-scale software development. This is based on the fact that the three main browsers were developed in rather different ways. However, this is somewhat speculative, and additional work is needed.

One major question is the comparison of open source and proprietary software development. Our results regarding Firefox and Internet Explorer provide some evidence for the potential superiority of large-scale open-source projects. Up to 2009 Firefox was quickly gaining market share at the expense of Internet Explorer, and our benchmark results indicate that it appears to have had superior performance for most of them (this conclusion is restricted, however, by the fact that we did not measure Internet Explorer 6 and 7 and the early versions of Firefox). It also appears to have been more innovative, as reflected by having some "wins" in early introduction of new features, and much less "losses" than Internet Explorer. This is an important result, as it demonstrates that a large open-source project can in fact prioritize features better than a competing product developed in-house by a leading software firm. Of course this does not imply that this is always the case, but it provides an important case study as an instance.

However, in later years Chrome came to overshadow Firefox. To the degree that Chrome is an in-house product this implies that large company projects can also be better than open-source ones. The conclusion would then be that the main factor is not the project management style but rather the companies involved, in this case Microsoft as opposed to Google. But such a conclusion is tainted by the fact that Chrome is closely related to the open-source Chromium project. So maybe the most important factor is the various project managers and contributors. This calls for further investigation as noted in the future work section below.

Another sometimes contentious aspect of software development is the use of agile methodologies with a rapid release cycle as opposed to heavier plan-based methodologies with large-scale infrequent releases. Tabulating the browser version release dates indicates that Chrome and Firefox transitioned to rapid development methods, releasing a new version every 4–8 weeks (Fig. 2). This meant that there were more releases and each release contained fewer new features, leading to more focus in the work on each new release. At the same time, with rapid releases the development teams could respond more quickly to their competitors' released features which they considered to be important, and also respond quickly to user feedback and requests. Microsoft retained the traditional slow release cycle for Internet Explorer, releasing only 4 versions during the 5 years of the study, compared with 31 released versions of Chrome. This may have contributed to Internet Explorer's downfall.

## Threats to validity

Various threats to validity have been mentioned in previous sections. Here we note them together and expand on them.

The first threat relates to the assessment that Chrome is the dominant browser, as shown in Fig. 1. First, as noted, this data comes from StatCounter, and other counting services may reach different conclusions. Second, we focused on desktop systems, and the picture may be different on other platforms such as mobile. To address these concerns we checked other counting services and platforms, and found that most of them indicate that Chrome is of growing importance and often dominant. The most prominent dissenter is

netmarketshare.com, which claims that Internet Explorer is still the dominant browser worldwide by a large margin (58% for Explorer vs. 23% for Chrome in January 2015) (http://netmarketshare.com/). The difference is probably due to a much smaller sample (40 thousand web sites as opposed to 3 million for StatCounter) and differences in methodology, including an attempt to count unique users per day and to weight countries by their total traffic. We believe that the StatCounter data is more reliable, and specifically prefer to count activity and not users. The issue of the mobile market is mentioned below in the future work section.

Another threat to the validity of the work reported so far is its focus on purely technical aspects of browsers. We did not check the marketing aspect of the browsers, hence, we cannot separate the technical superiority from the brand name. For example, according to Dave Parrack (http://www.makeuseof.com/tag/7-awesome-google-chrome-promo-videos/) an important aspect of Chrome's rise was "the great promotional efforts produced by Google" in the shape of promotional videos released on the web. Examples of 7 such videos are given, including the "Chrome speed tests" video released in May 2010 that went viral; at this time Chrome was just beginning its rise in market share, and the video may have contributed to its momentum. All 7 videos were released by April 2012, when Chrome had already overtaken Firefox but was still second to Internet Explorer. Additional work from a marketing perspective is needed to alleviate this concern.

A third threat is that we compared Chrome's performance with only two main rivals (Internet Explorer and Firefox) and partially also with a third (Opera). There are many other browsers as well, and maybe Chrome is not better than all of them. The focus on this set of contenders is justified by the fact that together with Chrome they account for well over 90% of the desktop browsers market. However, if any of the smaller browsers is indeed superior to Chrome and the others, this would testify to the importance of branding and marketing relative to technical considerations.

Using existing benchmarks is also a threat to validity, especially since their documentation is sometimes short on details of exactly what they measure and how. However, it should be noted that benchmarking browsers (and other system types for that matter) is not trivial. Therefore we preferred to rely on prominent benchmarks that have established themselves over the years instead of trying to devise our own—and risk threats to validity that result from our inexperience in such benchmarking. That being said, it should be noted that the benchmarks do not test all possible aspects of browser technology. For example, it is possible to conduct a more detailed study of compatibility issues (*Chaudhary, Prasad & Orso, 2013*) to try and quantify the problems that may occur with each browser.

Finally, a possible threat to validity concerning the introduction of new features is that such features could be introduced in plugins before being integrated into the browser core. This would cause the dates of the releases which first included these features to be misleading. However, we do not consider this to be a serious threat as even the most popular plugins are used by only a small fraction of users.

**Future work**

A drawback of the current work is its focus on the desktop market. Obviously examining competing browsers in the mobile market would also be interesting. Using StatCounter.com data, it turns out that Chrome is now also the leading browser on mobile platforms (http://gs.statcounter.com/). However, its rise started much later, and accelerated considerably only in 2014, eventually surpassing both the Android and Safari browsers. It would be interesting to repeat our measurements with multiple releases of these browsers, and perhaps also with UC Browser, which is the fourth-ranked browser and also seems to be gaining market share, especially in emerging markets.

Another potentially interesting line of study is to try and compare the relative importance of technical considerations, marketing campaigns and practices, and brand name. It is widely accepted that Internet Explorer gained its market dominance by being bundled with the Windows operating system, and it is reasonable to assume that the strength of UC Browser in emerging markets is related to the strength of the company which developed it, Chinese mobile Internet company UC Mobile. Chrome most probably benefited from the Google brand name and from Google's marketing campaign. But how to separate these effects remains an open question.

In the late 1990s the browser war between Internet Explorer and Mozilla (later Firefox) was portrayed in colors of a race between proprietary software and open source software. Chrome is a unique combination of both. It was initially developed within Google, but then it was largely turned into an open source project. An open question is whether it was really turned over to the open source community, or remains largely under Google control, both in terms of code contributions and in terms of management. Thus an interesting direction for further work is to dissect the sources of advances made in Chrome (or rather Chromium), and to see how many of them can be attributed to developers outside Google.

## CONCLUSIONS

We tested the technical performance of the three major browsers (Chrome, Firefox, and Internet Explorer) and compared the release times of 25 features. Overall it seems that all three browsers became better over time, as most of the benchmarks that were examined showed a clear improvement trend, and all the browsers evolved and received better results. It is also apparent that the release rate of versions became more frequent over the years (especially for Chrome and Firefox).

In conclusion, the cumulative evidence we have collected indicates that Chrome's rise to dominance is indeed consistent with technical superiority over its rivals and with insightful management of feature selection. However, we still cannot say that it is *the result* of technical superiority alone, as marketing and the Google brand probably also played an important role. Studying the marketing campaign may well be a worthwhile effort.

## ACKNOWLEDGEMENTS

Many thanks to Zvi Gilula for his explanations about statistical procedures and for providing the software used in the analysis. This work is a greatly extended version of a

class project done with Kobi Atiya, who contributed to the conceptual design and initial results. The Opera measurements were performed by Amir Massarwi.

### Funding

This work was not specifically funded by any agency.

### Competing Interests

Dror Feitelson is an Academic Editor for PeerJ.

### Author Contributions

- Jonathan Tamary conceived and designed the experiments, performed the experiments, analyzed the data, wrote the paper, prepared figures and/or tables, performed the computation work, and reviewed drafts of the paper.
- Dror G. Feitelson conceived and designed the experiments, analyzed the data, wrote the paper, and reviewed drafts of the paper.

### Supplemental Information

Supplemental information for this article can be found online at http://dx.doi.org/10.7717/peerj-cs.28#supplemental-information.

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
