# Peer review of "The rise of Chrome"

_PeerJ Computer Science, doi:10.7717/peerj-cs.28_

## Round 0.1 · original submission · Major Revisions

Your paper addresses an interesting topic that is relevant for computer science community. Unfortunately the experimental design of the study is poor as well as the discussion of the experimental results.

If you intend to submit a new version of the paper, you should improve the design of the experiment according to the comments and suggestions provided by both reviewers.

In particular I suggest you present your study in a more structured way, according to well-known guidelines, such as the ones provided by:
Jedlitschka, A.; Pfahl, D., "Reporting guidelines for controlled experiments in software engineering," Empirical Software Engineering, 2005. 2005 International Symposium on , vol., no., pp.10 pp.,, 17-18 Nov. 2005

Reviewer 1 ·

Basic reporting

The paper reports a study for the comparison of three of the most popular browsers for desktop PCs: Google Chrome, Internet Explorer and Mozilla Firefox by considering the versions released in the last 6 years. Three factors have been considered in three separated parts of the sudy:
- The scores obtained by executing well-known benchmarks;
- The measured start times;
- The number of features provided by each browser, in a set of 25 relevant features selected by the authors. To measure the relative importance of these features, the authors proposed a complex statistic approach based on the judgements of 254 users.
A brief discussion that tries to obtain some lesson learned is placed in the last paragraphs.

Experimental design

The authors measured the benchmark scores for any release of any of the three browsers and reported the results in figures. No further comparative analysis on these data are reported.
The complete set of parameters considered by these benchmarks is not reported, so I am not able to know, in example, if and how parameters such as the memory usage has been taken into account and in what measure. The confidence in the experimental data depends on the validity of the benchmarks, that is not discussed in this paper.
As regards the third part of the experimentation, the selected features are analyzed in order to evaluate only wins (i.e. features that a browser provided early with respect to the other ones) and losses (i.e. features that a browser provided later with respect to the other ones): it seems a quite reductive analysis. On the other hand, a complex statistical approach has been adopted to weigh the importance of the features for the users. The adopted methodology could be described in a more formalized and detailed way, giving space to the threats to construct validity related to this methodology.
In conclusion, the experimental design lacks of:
- A more detailed description of the set of aspects that the benchmark takes into account;
- A discussion of the threats to the validity;
- A more convincing explanation of why the set of three experiments that have been carried out is able to provide a complete judgement regarding the user perceived quality of the three browsers.

Validity of the findings

As pointed out in the Experimental Design section, the lack of a Threats to Validity subsection is a great limitation with respect to the assessment of the validity of the findings.
The insight provided in the discussion sections (5.2 and 5.3) are intuitively acceptable but they are not inferred from the data.
I think that the experiment is interesting for historical reasons but its contribution in terms of lesson learned is very uncertain and limited.

Additional comments

Section 2.2 is not of great interest: in this section data from StatCounter.com are cited to show the use of its browser in recent years, but without relevance with respect to the other analysis reported in the rest of the paper.
I do not find the “Insight for R” software package: please give a reference for it.
The Opera browser is introduced only in section 5.2. Why it has not been considered in the rest of the analysis?
As regards the third part of the experimentation, a historical evaluation of the score of the features provided by the browsers over time could be reported, on the basis of the evaluated importance scores.
Modern browsers offer the possibility of being integrated with plug-ins, which often are widespread, as in the case of AdBlock. The presence of these plug-ins is able to threat the validity of some of the “wins” and “losses”, since in many cases you consider a browser as not having a feature when it was actually available even before it was natively incorporated into the browser.
The study is quite interesting but it is only able to give a view of the history of three of the most popular browsers, but if fails in terms of lesson learned that can be useful for the future. The findings provided in the discussion sections (5.2 and 5.3) are intuitively acceptable but they are not inferred from the described experimentation.

Cite this review as

Reviewer 2 ·

Basic reporting

This paper proposes a study of performances and feature from most widespread browsers. The performances are evaluated by means of available benchmarks whereas
the feature set was based on a survey on the importance of 25 major features.
The structure of the paper is appropriate.

Experimental design

From the point of view of experimental design i think that both section 3 (technical performance assessment) and 4 ( feature selection) should be improved.

In section 3 the performance assessment is performed only using javascript benchmarks and none is said for a lot of important aspects (machine setups, configuration of nodes with respect to Operating System running the benchmark and so on).
There are a lot of aspect that are not even taken into account:
- caching efficiency
- page reloads
just to name few. My opinion moreover is that this section needs an improvement to study the influence on the proposed results of external factors that influence the provided results.

For what regards section 4 the feature selection i think that the classification of feature is too coarse. The survey it should be structured taking into account kind of user and applications in order to better qualify the scopes of browsers usage by classes of users.

Validity of the findings

In general i think that the overall structure of the paper is poor from a technical point of view: there are no research questions that are formulated and the analysis is only performed to gather data without any research statement behnind. My hint to authors is to improve on this: what research questions they are trying to answer? what is the final goal of the paper ? This is not clear, if the goal is to assess that chrome has become the most widespread browser due to technical superiority on both performance and feature sides the paper should be better restructured highlighting adequately the research questions and taking into account marketing issues (that are informally and qualitatively discussed but have a great impact on the final browser adoption statistics)

Additional comments

No comment.

Cite this review as

---

## Round 0.2 · accepted · Accept

The paper was modified according to most of the reviewers comments and suggestions. The aim of the study, its research questions, the way you addressed them, the results and the conclusions are now better presented. According to the PeerJ editorial guidelines, it is my opinion that the paper is now acceptable for publication.